# The Challenge of Diffusion Magnetic Resonance Imaging in Cerebral Palsy: A Proposed Method to Identify White Matter Pathways

**DOI:** 10.3390/brainsci13101386

**Published:** 2023-09-29

**Authors:** Ophélie Martinie, Philippe Karan, Elodie Traverse, Catherine Mercier, Maxime Descoteaux, Maxime T. Robert

**Affiliations:** 1Centre for Interdisciplinary Research in Rehabilitation and Social Integration, Québec, QC G1M 2S8, Canada; ophelie.martinie.1@ulaval.ca (O.M.); elodie.traverse@cirris.ulaval.ca (E.T.); catherine.mercier@rea.ulaval.ca (C.M.); 2Department of Rehabilitation, Université Laval, Québec, QC G1V 0A6, Canada; 3Department of Computer Sciences, Université de Sherbrooke, Sherbrooke, QC J1K 2R1, Canada; philippe.karan@usherbrooke.ca (P.K.); maxime.descoteaux@usherbrooke.ca (M.D.)

**Keywords:** diffusion neuroimaging, cerebral palsy, tractography

## Abstract

Cerebral palsy (CP), a neuromotor disorder characterized by prenatal brain lesions, leads to white matter alterations and sensorimotor deficits. However, the CP-related diffusion neuroimaging literature lacks rigorous and consensual methodology for preprocessing and analyzing data due to methodological challenges caused by the lesion extent. Advanced methods are available to reconstruct diffusion signals and can update current advances in CP. Our study demonstrates the feasibility of analyzing diffusion CP data using a standardized and open-source pipeline. Eight children with CP (8–12 years old) underwent a single diffusion magnetic resonance imaging (MRI) session on a 3T scanner (Achieva 3.0T (TX), Philips Healthcare Medical Systems, Best, The Netherlands). Exclusion criteria were contraindication to MRI and claustrophobia. Anatomical and diffusion images were acquired. Data were corrected and analyzed using Tractoflow 2.3.0 version, an open-source and robust tool. The tracts were extracted with customized procedures based on existing atlases and freely accessed standardized libraries (ANTs, Scilpy). DTI, CSD, and NODDI metrics were computed for each tract. Despite lesion heterogeneity and size, we successfully reconstructed major pathways, except for a participant with a larger lesion. Our results highlight the feasibility of identifying and quantifying subtle white matter pathways. Ultimately, this will increase our understanding of the clinical symptoms to provide precision medicine and optimize rehabilitation.

## 1. Introduction

Cerebral palsy (CP) is the most common neuromotor disorder in children [1,2]. It is a consequence of the various non-progressive brain insults occurring at various stages of the developing fetal or infant brain, such as brain infections, periventricular leukomalacia, malformations, or perinatal strokes [3]. The heterogenous disturbances of the early brain lead to highly variable lesion sizes, which have a direct impact on the gray and white matter integrity across all regions of the brain and central nervous system pathways [4]. Associations have been found between sensorimotor impairments and structural brain lesions [5,6] as well as with the integrity of white matter tracts [4,7]. While there is a consensus among researchers and healthcare policymakers [8] on the need to further investigate the relations between brain structure and overall functions as well as brain reorganization during development in the CP population, several methodological challenges remain.

Originally, neuroimaging modalities such as magnetic resonance imaging (MRI) and computed tomography (CT) were used mainly to diagnose CP [9]. Over the last two decades, more advanced tools have been used, such as diffusion-weighted imaging, to characterize brain damage [4]. Diffusion MRI is the only method allowing a non-invasive in vivo study of white matter organization (dMRI; [10]). It implicitly represents white matter tract organization by modeling water molecules’ coherent motion determined by physical and biological obstacles along these tracts [11,12]. One of the dMRI models to reconstruct these coherent tracts is diffusion tensor imaging (DTI; [13]). DTI has been widely used in children with CP to quantify white matter integrity in the whole brain [14] and several pathways including, but not limited to, corticospinal tracts [15,16], corpus callosum [6,7], and thalamocortical radiations [17]. Some studies using DTI measures showed improvement in brain integrity following intensive rehabilitation, which suggests neuronal plasticity [6,15,18], while others observed no modification [19,20]. The discrepancies between the results from a handful of studies reinforce the need for more robust methods to increase the validity of results and decrease heterogeneity across studies.

A common limitation in most studies conducted in CP is the absence or partial information provided about the preprocessing and signal reconstruction steps. As a result, it is difficult to assess the quality of the methodology and to compare results across studies. Indeed, a recent study showed inconsistencies in results between DTI metrics in CP [21], highlighting the importance of choosing appropriate measures to improve psychometric properties (e.g., sensitivity and reliability) and to increase finding robustness [22,23,24]. A key challenge is that DTI can only compute one fiber direction within a voxel [25], a considerable issue given that 60 to 90% of white matter voxels contain multi-fibers [26]. To address this limitation, new local reconstruction methods emerged to solve bending, fanning, kissing, and crossing fibers issues and to improve the biological accuracy of fiber estimation [26,27,28]. For example, the constrained spherical deconvolution model (CSD) allows the study of more subtle pathways, such as the corticocerebellar system (i.e., corticopontine and cerebello-thalamo-cortical pathways) [29,30,31,32]. Additionally, multi-compartment methods such as neurite orientation dispersion and density imaging have emerged to characterize the microstructural complexity of white matter tracts [33] and to better interpret DTI observations [11]. Such advances can thus pave the way for the investigation of new pathways and functions in CP. Thus, it is timely to update current results with newer and more robust models [4].

Aside from the methodological challenges that are common to various applications of DTI, neuroimaging in children with CP presents specific challenges due to the difficulty of segmenting brain tissues with extensive lesions that show abnormal signal intensity or anatomical shape [34]. These challenges come from the wide range of etiology and variations in lesion size. While some children may exhibit a “hole” in their brain, others may have subtle cortical or subcortical malformations. These variations pose challenges for some methods that rely on a priori assumptions about brain structure [35]. As a result, children with CP who present large lesions are often excluded from analysis [36]. To better represent CP population heterogeneity [37,38], researchers must agree on a transparent methodology to process brain data in individuals with CP.

The overall goal of this paper is to demonstrate the feasibility of using a standardized and open-source pipeline for neuroimaging analyses of children with CP presenting heterogeneous lesions. Preprocessing, computing fiber orientation, extracting tractograms, and diffusion metrics were applied to major sensorimotor pathways (i.e., corticospinal, corpus callosum, and mediolemniscal tracts) and premotor pathways of the corticocerebellar system (i.e., fronto-ponto-cerebellar and cerebello-thalamo-frontal tracts). 

## 2. Materials and Methods

### 2.1. Data Collection

A total of 8 children were recruited. The inclusion criteria were (1) a diagnosis of cerebral palsy with spastic hemiplegia, (2) being aged between 8 and 17 years old, and (3) eligibility for MRI scans. Exclusion criteria were (1) ferromagnetic neuroimplants or presence of metal in the head, (2) other neuroimaging contraindications, (3) non-controlled epilepsy, and (4) inability to perform MRI scans. All participants and their legal tutor gave their consent. This research was approved by the Centre intégré universitaire de santé et de services sociaux de la Capitale-Nationale ethics committee (ethic’s number: RIS_2021-2034).

Participants were invited to a single MRI session. The following method was chosen according to the processing recommendation [39,40], considering software and methods limitations [41], and parameters were adapted to our CP data.

### 2.2. Data Acquisition

The neuroimaging data were acquired on a 3T Philips Scanner (Achieva 3.0T (TX), Philips Healthcare Medical Systems, Best, The Netherlands) with a 15-channel head coil at Synase Clinic in Quebec City (Quebec, QC, Canada), without any sedation. The total time of data acquisition was 21 min. To prepare the children for the MRI session, a short, age-appropriate explanation was provided on how the session would occur. In addition, a video and audio recording of the MRI procedure were shown. A structural T1-weighted image was acquired with a turbo field echo multishot protocol in the axial plane (voxel size = 1 mm isotropic, 180 slices, TR = 7.3 ms, TE = 3.3 ms, inversion time = 940 ms, flip angle = 9°, and duration = 6 min) to perform anatomical registration. Diffusion MRI was acquired using an Echo-Planar Imaging single shot protocol in 60 slices with a HARDI sequence (protocol detailed in https://zenodo.org/record/2602049 (accessed on 5 June 2023); slice thickness = 2 mm, matrix 112 × 112, in-plane resolution = 2 × 2 mm, flip angle = 90°, FOV = 224 × 224 mm, 84 directions, b-values = 300 s/mm^2^, 1000 s/mm^2^ and 2000 s/mm^2^, 6 b = 0 volumes, TR = 7650 ms, TE = 89.5 ms, and duration ~16 min). This sequence was chosen because the multi-shell acquisition is compatible with NODDI analysis, but a trade-off with a number of directions was found to not increase too much acquisition time because of the hardware limitations of the system. Moreover, this protocol has good reliability [42]. A reversed encoding non-diffusion sequence was also acquired to improve field inhomogeneity correction.

### 2.3. Data Preprocessing

All the preprocessing and processing steps were performed on computational resources offered by Calcul Quebec (https://www.calculquebec.ca/, accessed on 10 January 2023 ), a provincial partner of the Digital Research Alliance of Canada (Quebec, Canada) (https://alliancecan.ca/en, accessed on 10 January 2023).

#### 2.3.1. Data Correction

Data preprocessing and correction were performed with the Tractoflow version 2.3.0 pipeline (code available at https://github.com/scilus/tractoflow.git [40]), which is based on other existing tools such as Diffusion Imaging for Python library version 1.5.0 (DIPY; [43]), FMRIB software library 6.0 (FSL; http://fsl.fmrib.ox.ac.uk/fsl/fslwiki/, accessed on 1 September 2022; Oxford, UK; [44]), MRtrix3 (https://www.mrtrix.org/, accessed on 1 September 2022; [45]), Advanced Normalization Toolbox 2.4.1 [46] and containers such as Nextflow and Singularity to improve reproducibility [47,48]. The -bundling profile option was used for its robustness with challenging data such as big lesions, which usually hinder the T1 segmentation. This option consists of activating tailored settings for tracking. Since the script was launched on a high-performance computer, the -fully_reproducible option was also set to maximize the reproducibility of the results by allowing multi-thread task execution without diminishing analysis quality. Both options were activated for all participants. For a graphical representation of the overall pipeline, see Figure 1.

The T1 and dMRI preprocessing steps included brain extraction, denoising, motion and field inhomogeneity correction, normalization, resampling in parallel to T1 registration, and segmentation.

#### 2.3.2. Diffusion Metrics

Diffusion metric computation was performed within the Tractoflow pipeline. Since multi-shells have been shown to be more sensitive for DTI analysis in children [49], the DTI shells (i.e., 0, 300, 1000 s/mm^2^) were extracted to compute the DTI metrics (i.e., fractional anisotropy (FA), axial diffusivity (AD), radial diffusivity (RD), and mean diffusivity (MD)) using DIPY [43]. The fiber orientation distribution function (fODF) shells (i.e., 0, 1000, and 2000 s/mm^2^) were extracted to compute fODF measure (i.e., AFD total, 50; NuFO, 51; and fixel AFD) with a maximal spherical harmonic order of eight, based on using constrained spherical deconvolution (CSD) [26]. AFD is known to correspond to the amplitude of fODF [25]. Thus, AFD total indicates the overall density in a voxel, while fixel AFD indicates the AFD of a specific subset of the fiber population within a voxel [50]. NuFO is a metric that represents the number of fiber orientations within a voxel based on the number of local maxima of the fODF [51]. In addition, neurite orientation dispersion and density imaging (NODDI) metrics, a multi-compartment model [33], were computed from the resample DWI, the resample b0, and the bval and bvec files from the eddy/topup correction steps, with noddi_flow (code available at https://github.com/scilus/noddi_flow.git, accessed on 9 May 2023 [52]).

#### 2.3.3. Whole Brain Tractography

An ensemble tractography [53] strategy was used with the -bundling profile. Probabilistic tracking was performed with Particle Filter Tracking option on fODF in T1-based white matter map to reconstruct whole brain tractography with default parameters (FA threshold = 0.1, step size = 0.5 mm, maximum angle between 2 steps = 20°, number of seeds per voxel = 10, and stopping criteria = gray matter mask). In addition, the local tracking method was also performed, which relies on local orientation within each voxel. This tracking method has been shown to have efficient tracking capabilities despite large lesions and, thus, maximizes the chance of capturing coherent fibers’ direction, even if they do not reach gray matter [54]. For local tracking, default parameters were used (FA threshold = 0.1, step size = 0.5 mm, maximum angle between 2 steps = 20°, and number of seeds per voxel = 10). However, for Subject 6 (S6), we only used local tracking since PFT tracking is based on anatomical images, and this participant showed abnormal brain structure, which led to unusable white/gray/cerebrospinal fluid masks. A visual inspection was performed at each step to check the quality of the data and the output of each preprocessing step.

#### 2.3.4. Tractogram Extraction

Two different procedures were conducted. An automatic extraction was performed for both the corticospinal tracts (CSTs) and the corpus callosum (CC). For the mediolemniscal (ML) pathways, the fronto-ponto-cerebellar (FPC) tract, and the cerebello-thalamo-frontal (CTF) tract, an atlas-based tailored procedure was used. Given that this segmentation method relies on a priori anatomical knowledge of and is sensitive to image contrast and intensity [55], factors that are known to be hindered in cerebral palsy [34], the parameters were fine-tuned to obtain maximal deformation of the atlas. This adjustment was made to ensure the fitting of the atlas to our participant’s anatomical spaces.

##### Automatic Extraction

For extracting well-known main bundles such as CSTs and CC, a reproducible, robust, and multi-atlas method called RecobundleX was used (code available at https://github.com/scilus/rbx_flow.git [56,57]) with the -fully_reproducible option. This virtual dissection method allows the extraction of 39 bundles according to the population average and tractoflow output (i.e., FA map and a combination of local and PFT tracking maps). Only streamlines between 50 and 250 mm were extracted. CC was extracted in 6 different parts depending on the projection cortex (i.e., two frontal parts, one pre- and post-central gyri part, one parietal, one occipital, and one temporal part) in the RecobundleX workflow.

##### Atlas-Based Customized Extraction

For subtle bundle extraction, such as for the ML and the frontal pathways of the corticocerebellar system (i.e., FPC and CTF), we used the procedure described in the section below.

Atlas registration

Regions of interest (ROIs) were found from atlases. We used the JHU ICBM DWI 1mm atlas from the John Hopkins University DTI-based white matter atlases (available at https://neurovault.org/media/images/264/JHU-ICBM-DWI-1mm.nii.gz [58]) to extract anterior and posterior limb of the internal capsules, medial lemniscus, right and left superior cerebellar peduncles (SCPs), and middle cerebellar peduncle (MCP). As contralateral MCP is more effective in reconstructing frontopontine tracts [31], this region was manually separated into two sides following the interhemispheric axis using MI-Brain [59]. Ventral, lateral, and posterior nuclei of both thalami were extracted from the Talairach atlas (available at http://www.talairach.org/ [60,61]). Premotor cortices and postcentral gyri were extracted from the Brainnetome atlas (available at http://www.brainnetome.org/resource/). The transformation of atlases and ROIs to our participants’ anatomical space was performed using the ANTs toolbox. The anatomical images from atlases in the MNI space were converted into our participants’ anatomical space. The transformation matrix obtained following this procedure was subsequently utilized to perform a transformation that deformed ROIs from the atlases to align with the anatomy of our participants. We visually verified the ROI deformation to ensure that the transformation was correctly performed and followed our participants’ anatomy. See Figure 2 for a comparison between the healthy and lesioned sides for all ROIs. 

The detailed registration procedure and its parameters are available in Appendix B.

Since the registration failed for S6 because of hemisphere proportion deformation due to the lesion, leading to an untrusting ROI location in this participant’s space, the atlas-based extraction method was not possible. For the misregistration of ROIs, see Figure 3.

2.ROI and Bundle Extraction

The following procedures were performed using the Sherbrooke Connectivity Imaging Lab Python (SCILPY) dMRI processing toolbox (codes available at https://github.com/scilus/scilpy.git). ROIs were then extracted from the registered and transformed atlases using the scil_split_volume_by_labels.py, and an atlas with all ROIs was reconstructed for each participant with scil_combine_labels.py [62]. For a visual representation of the ROIs for ML, FPC, and CTF tracts, see Figure 4. 

The tractograms were extracted with scil_filter_tractogram.py between our ROIs from en-semble tracking, using “any” and “include” parameters. For FPC tracts, ROI 1 was the premotor cortex, ROI 2 was the homolateral cerebral peduncle, and ROI 3 was the contra-lateral MCP. We excluded tracts passing through the CC and/or the contralateral internal capsule. For CTF tracts, ROI 1 was the SCP, ROI 2 was a fusion of ventral and lateral nuclei of the contralateral thalamus, and ROI 3 was the contralateral premotor cortex. We ex-cluded tracts ending in motor and sensorimotor cortices. To extract ML pathways, ROI 1 was the medial lemniscus, ROI 2 was the ipsilateral ventral posterior lateral nucleus of the thalamus, and ROI 3 was the ipsilateral postcentral gyrus.

Before the analysis, tractograms were cut with scil_cut_streamlines.py to remove the outgoing streamlines outside our ROIs, scil_detect_streamlines_loops.py [63] to avoid loops in our tracts and cleaned with scil_remove_invalid_streamlines.py. Final cleaning of spurious streamlines was performed using scil_filter_streamlines_by_length.py with a threshold set at 120 mm to eliminate streamlines with impossible anatomical pathways.

### 2.4. Lesion Characterization

As some participants did not show a clear separation between lesion volume and cerebral tissues (e.g., dilated ventricles), and because of the heterogeneity of lesion type in our sample, the lesion was characterized by volume ratio between gray and white matter for each hemisphere [64]: (gray + white matter) left/(gray + white matter) right. A ratio near 0 indicates an important difference in tissue volumes between hemispheres, whereas a score near 1 indicates equal tissue volume between hemispheres. To segment tissues for each hemisphere, the Freesurfer segmentation method (code freely available at http://surfer.nmr.mgh.harvard.edu/ [65]) was used with a freesurfer_flow (code available at https://github.com/scilus/freesurfer_flow) on registered T1 outputted from Tractoflow. In addition, as children with hemiplegic CP have lesions mostly located in one hemisphere [66], an asymmetry index (AI) for volume was computed between the contralesional (C) and the ipsilesional (I) hemisphere using the following formula: AI = (C − I)/(C + I) [67]. A positive ratio shows that the contralesional hemisphere has a bigger tissue volume than the ipsilesional one. A score near 0 indicates no differences between hemispheres.

### 2.5. Tracometry

Tractometry was then performed using tractometry_flow (code available at https://github.com/scilus/tractometry_flow.git) based on Nextflow from the SCILPY dMRI preprocessing toolbox [68] to give metrics by bundle output from RecobundlesX for CSTs and CC and from our tailored tracts extraction, for the ML, FPC and CTF tracts. The chosen metrics were the tract volume and the FA, MD, RD, and AD from DTI shells.

The metrics extracted from fODF shells are the fixel AFD, total angular fiber density (AFD), and the number of fiber orientations (NuFOs). Fixel AFD represents the main fODF peak direction [25], while AFD total represents all the peak directions within a voxel. AFD is recognized as an interesting metric in addition to FA since it is highly correlated with FA but less sensitive to noise and more efficient in the presence of crossing fibers [69]. The NuFO measure is computed from the number of local peaks [51].

NODDI metrics are the neurite density index (NDI), the orientation dispersion index (OD), and the isotropic compartment (ISO) [33]. The NDI represents the extraneurite compartment within a voxel, corresponding to the extracellular volume fraction. The OD is the orientation distribution of the cylinder-shaped fibers from 0 (i.e., perfectly aligned fibers) to 1 (i.e., isotropic). The ISO corresponds to the freewater compartment. As this microstructural imaging model has greater sensitivity than FA, it is a promising method to better describe intravoxel diffusion signals in clinical population [70].

An AI was also computed for each of the previously described variables to highlight differences between the ipsilesional and the contralesional hemispheres.

## 3. Results

### 3.1. Clinical Data

Eight participants (four girls and four boys) aged between 8 and 13 years old (M = 11.4, SD = 1.6) had MRI scans. Six had CP due to pre- or perinatal strokes, one caused by periventricular leukomalacia, and one had an unknown etiology. The children had a level I to III on the Manual Ability Classification System (MACS), corresponding to light to moderate difficulties for a child with CP to handle objects in daily activities. For a demographic description of the participants, see Table 1. Participants S6 and S8 underwent two scanning sessions because of important movements during acquisition. No participants were excluded due to poor image quality. DMRI encoding was successfully performed.

The freesurfer_flow was able to segment tissues for six participants (i.e., S1, S2, S4, S5, S7, S8). This step failed for participants S3 and S6 because of the large lesion size, the deformation of the structure of tissues, and the abnormally enlarged proportion of the non-lesioned hemisphere. Hence, lesion extent cannot be quantitatively expressed for S3 and S6, but the lesion volume was bigger than the other participant’s lesion. See Table 2 for a summary of brain volumes and tissue ratio and Figure 5 for a visual representation of the T1 image.

### 3.2. Automatic Tract Extraction

Out of the 39 possible tracts allowed by RecobundleX, only the left and right inferior cerebellar peduncle were not extracted because the field of view cropped the lower part of the brainstem. This was a trade-off decided during acquisition to not lengthen the protocol and because we had no particular interest in those tracts. The additional tracts are displayed in a mosaic in Appendix A. Out of the total 37 possible tracts, 31.6% were extracted for all the participants regardless of the etiology of the lesion (see Appendix A and Appendix A in Appendix A for the total description of the tracts extracted with this procedure). Some of the tracts were missing for some participants due to lesion location (e.g., right frontal aslant tract for S3 or all corpus callosum tracts for S6).

For this study, only the CSTs and CC are presented in the automatic tract extraction section below because of the interests of the neuroimaging community working with the CP population.

#### 3.2.1. Corticospinal Tracts

CSTs were reconstructed for all participants (see Figure 6). 

All participants presented both right and left CSTs except S6, whose right CST was missing due to a large lesion extending to almost the whole volume of the right hemisphere. Within each tract, streamlines were missing in accordance with lesion location and extent for all participants. For instance, S3 showed clear missing streamlines within the right lateral projections of the CST, probably due to the large lesion affecting mostly the lateral part of the motor cortex. All participants but S4 showed clear asymmetry in tract volume and diffusion metrics. The almost perfect symmetry observed for S4 might be associated with its clinical profile (i.e., minimal sensorimotor deficits) and the unknown etiology.

The difference between left and right volume seems to be a more sensitive metric to asymmetry in comparison to other metrics. For example, S2 clearly has thinner left tracts than the right one (respectively 33 cm^3^ and 74 cm^3^) but still presents an FA value of 0.46 for the left CST and 0.53 for the right CST. This observation is also similar for S3 and S5.

Moreover, the metrics extracted are consistent with each other. As expected, we observed a higher FA value for bigger tracts with lower MD values. These results were similar across all participants except for S3, who showed a high MD associated with a high FA value for the CSTs located in the lesioned hemisphere.

#### 3.2.2. Corpus Callosum

All six regions of the CC were identified for the participants except S6, for whom the CC tracts were not possible to extract (see Figure 7). 

The CC segment projecting to the posterior part of the frontal lobe (i.e., CC_Fr_2) was missing for S3 and partially for S8. This lack of streamlines might once again be caused by the lesion, mainly located on the posterior part of the frontal cortex for S3 and in the right hemisphere for S6, whereas S8 showed an enlarged ventricle. Lesions can be seen in Figure 4 in the previous results section.

As previously described, the volume metric seems to be a better indicator of the presence or absence of streamlines. This is particularly noticeable for the temporal part of the CC, which is clearly visible for S3 and S4 but more subtle for S1, S5, and S8. However, FA, MD, AFD, and OD remain high for all participants despite their volume value. The occipital part of the CC has the higher FA for all except S8, and the lower FA value is for the posterior part of the frontal lobe for all except S4. Besides the temporal part of the CC, the metrics are coherent. AFD values share the same pattern as FA values, and high FA/AFD is associated with low MD.

### 3.3. Atlas-Based Tracts Extraction

An atlas-based tract extraction procedure was successfully performed on ML, FPC, and CTF.

#### 3.3.1. Medio-Lemniscal Tracts Extraction

Both left and right ML tracts were reconstructed for seven out of the eight participants (see Figure 8). 

Reconstruction of the ML tracts was not feasible for S6 due to a registration step failure. Asymmetry between hemispheres is visually less noticeable than the CSTs.

A greater asymmetry was observed for the contralesional hemisphere for all participants except for S4 and S5. An interesting observation is that S5’s lesion, located in the frontal lobe, has fewer sensorimotor projections of the ML tract. As previously described, S3 showed an abnormal pattern of diffusion metrics with higher FA and AFD on the lesioned side, unusually associated with high MD and low OD. A more lateral tract was observed for S3 in comparison to other participants.

#### 3.3.2. Fronto-Ponto-Cerebellar Tracts

Both right and left FPC tracts were extracted for all participants (see Figure 9) except for S2’s left hemisphere due to its lesion located in the ROI. All the metrics were also correctly assigned to the tracts, although only one streamline was extracted for one hemisphere for S5, which showed a left frontal lesion.

Asymmetry of volumes and diffusion metrics is less obvious than CC and CST section results as FPC tracts are not lateralized. Hence, some participants, including S3, S4, and S7, showed greater volume for the FPC located in the ipsilesional hemisphere.

#### 3.3.3. Cerebello-Thalamo-Frontal Tracts

Metrics for left and right CTF tracts were extracted for all participants (see Figure 10).

Comparing the different metrics, inconsistencies were revealed in the results of the CTF. S2 showed only little asymmetry of volume and FA, MD, and AFD metrics but had a more pronounced asymmetry for OD in comparison to the other participants. S7 revealed incoherent asymmetry with higher volume for the tract located in the lesion hemisphere, although the value differences remain low. In accordance with previous results, S3 and S4 showed an atypical pattern of diffusion metrics and asymmetry regarding their lesion side.

## 4. Discussion

This paper presented the steps to preprocess dMRI data, reconstruct local DTI, CSD, and NODDI metrics, and perform whole brain tractography for data collected in children with CP through robust and automatic tools. The feasibility of extracting most tracts in most participants was demonstrated with both a well-known automatic procedure for large bundles and a customized atlas-based method for subtle tracts. However, the customized method struggled with a participant with a larger lesion, which made tract extraction impossible in this case. Manual methods to find ROIs remain a possibility to correct registration errors for the brains with large lesions. Moreover, the tract pattern for S3 for both methods showed a more diagonal direction for the lesioned hemisphere with abnormally higher metrics compared to the contralesional side. This unusual tract profile might question our methodology or indicate tract reorganization within the lesioned hemisphere. A good way to explore this result more would be to segment the tracts into different portions and observe the diffusion metrics either at different points or along the tract. This was successfully developed in children [71] and tested in Parkinson’s disease patients [72] and seems a promising method to reveal new insights into white matter fiber organization. Tractometry was also applied successfully on all tracts, although the discrepancies between metrics highlight the importance of extracting different measures to correctly interpret diffusion properties. Our methodology offers an important solution for correcting and analyzing dMRI data in children with CP with available tools. However, one of the main challenges that remains is regarding the tissue and lesion segmentation. The general adoption of this proposed robust and reproducible pipeline by both researchers and clinicians can contribute to open science practice in CP studies. This will only be feasible through the standardization of tractography, enabling the pooling of data sets from different sources [40].

This pipeline was originally developed to analyze data from healthy subjects [40] and then customized to adult pathological populations [73]. It has now been successfully adapted to process brain data from children with CP. The challenge of tractography studies in children with CP is the heterogeneous lesion patterns and volumes. As a result, the interpretation of brain imaging remained previously limited and difficult to generalize to all individuals with CP [74,75]. Our results demonstrated the feasibility of each preprocessing step to clean and correct dMRI data in children with CP, which constitutes a critical feat [76], considering the abundance of artifacts in dMRI data [39]. The asymmetry in metrics for the majority of analyzed tracts found in our results was consistent with well-described phenomena in neuroimaging findings from children with hemiplegic CP [19,29,30,67,77]. The level of attention of our proposed pipeline can help reduce the volume of data in CP studies that get eliminated due to tractography difficulty. As a result, our proposed pipeline can analyze more data sets, contributing to a better representation of the heterogeneity of this clinical population.

One of the major advantages of our method is the use of three different and complementary signal reconstruction methods (i.e., DTI, CSD, and NODDI), which allows a more accurate and complete description of tract integrity and thus, avoids oversimplification of white matter tracts in children with CP [22]. In our results, we found that higher FA is mostly present when the MD is low, indicating, as expected, that the diffusion is anisotropic because of white matter in the voxel and the diffusivity perpendicular to the main direction is low, decreasing the MD value. In addition, AFD often shares the same pattern as the FA metric, whereas low OD is observed with a high FA. Taken together, the interpretation of NODDI and DTI metrics can give more information on the microstructure within a voxel since NODDI is known to be more sensitive to FA [78]. For instance, OD shows a strong negative correlation with FA [79] because low anisotropy in FA can be caused by more fiber dispersion within a voxel, which is expressed by a high OD. The resulting fODF metrics can overcome the limits encountered by DTI metrics, such as a low FA, which is often misinterpreted as an absence of fibers instead of possible crossing fibers [22]. Currently, while FA shares the same sensitivity as AFD total, its specificity is lower [69]. Tractography studies in the pediatric population require consideration for the methodology selected because of developmental changes in brain tissue characteristics [80], such as DTI metrics changing non-linearly during development [81]. With the combination of DTI, fODF, and NODDI metrics, white matter tracts are better eluded with less presence of confounding bias [22]. Moreover, our rigorously customized tract extraction method enables the study of new or decussating white matter pathways in children with CP. It can also enhance the knowledge of better-known tracts such as the corpus callosum, often tracked with only DTI, which limited its reconstruction to a thin inter-hemispheric mohawk [6,7], while our method allows the capture of more fibers.

Our proposed method can be used to standardize the neuroimaging literature on children with CP. Moreover, the use of a semi-automatic procedure with tailored parameters allows the prevention of possible biases normally found in manual procedures [82] and may reduce the exclusion of participants with CP in neuroimaging studies. As there is no consensus on the method to apply in neuroimaging data in CP, research teams currently use different tools and pipelines. The use of our suggested preprocessing method can significantly improve the accuracy and reliability of the data [22]. Moreover, the generalization of this proposed method allows the study of other white matter pathways that have been underexplored due to the use of less robust tools. However, the programming skills and methodological knowledge needed by such tools can be a hindrance to their use by clinicians. This highlights the need for multidisciplinary collaboration between clinical sciences (e.g., neurorehabilitation and radiology) and natural sciences and engineering (e.g., mathematics, physics, and computer sciences) to transfer the knowledge of informatics tools to clinicians. The use of a standardized method can help and accelerate open science and data sharing and bring significant benefits not only to researchers but also to individuals and families with CP. Further investigation of the feasibility of our pipeline can be accomplished by increasing the sample size and by including various types and lesion sizes. Future studies should investigate the relationship between the severity of sensorimotor impairments and white matter integrity.

## 5. Conclusions

In this paper, we tailored an existing diffusion MRI pipeline to children with CP who have heterogeneous lesions. This pipeline goes from raw diffusion MRI data through state-of-the-art preprocessing, fiber orientations computation, whole brain tractography, and diffusion metric extraction quantitatively applied to automatically reconstruct major sensorimotor pathways and premotor pathways of the corticocerebellar system. This powerful method of quantitative analysis of imaging data collected can contribute to improved diagnosis and treatment of CP as well as the understanding of its underlying mechanisms. Additionally, the proposed method can find potential application in other challenging populations, such as individuals with tumors or individuals who had a stroke.

## Figures and Tables

**Figure 1 brainsci-13-01386-f001:**
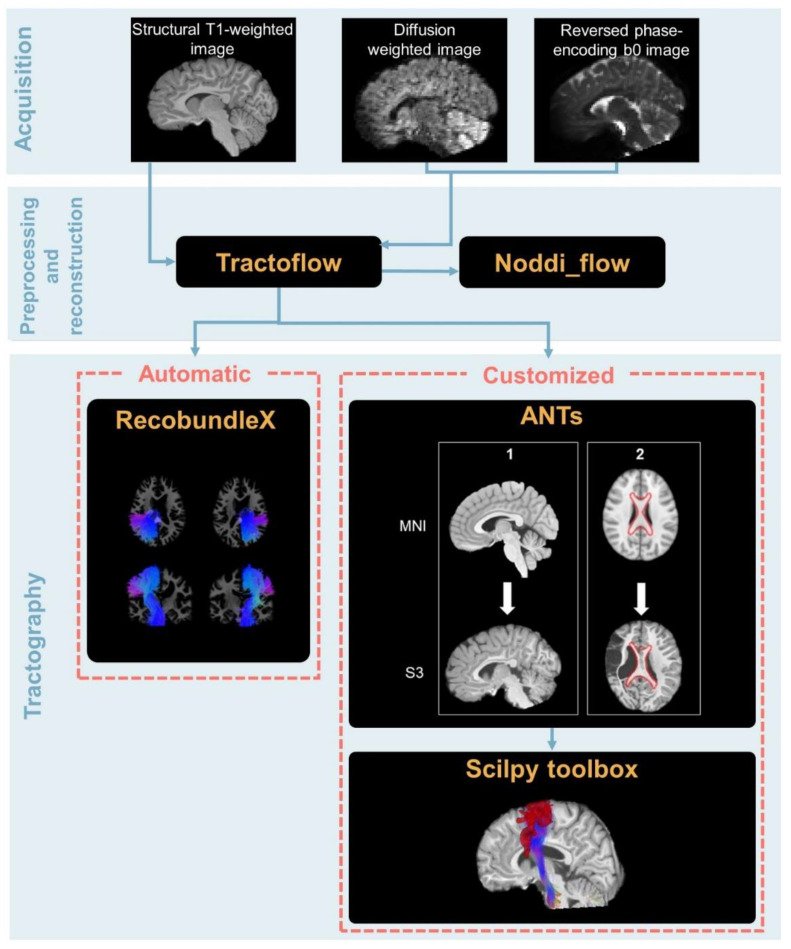
Overall pipeline from the acquisition to the tractography of the CSTs and CC (i.e., automatic procedure) and the mediolemniscal (ML) tracts, the FPC, and the CTF (i.e., customized atlas-based procedure). For the customized procedure, the first step (i.e., registration with ANTs) is divided into a first sub-step consisting of registration for the full betted and registered T1 and a second consisting of ROI registration. The tools used for each step are written in yellow ink.

**Figure 2 brainsci-13-01386-f002:**
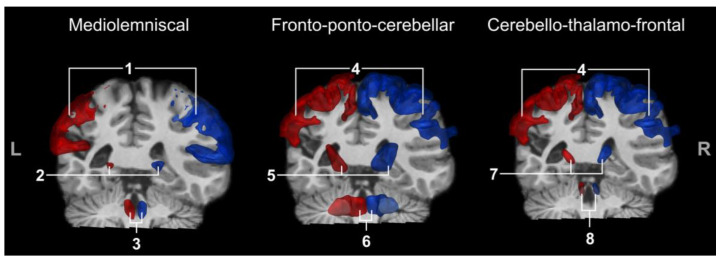
Lesioned (red) and healthy (blue) sides with all ROIs registered; 1: postcentral gyri, 2: ventral posterior lateral nuclei of thalami, 3: medial lemnisci, 4: premotor cortex, 5: internal capsule, 6: middle cerebellar peduncles, 7: ventral and lateral nuclei of thalami, and 8: superior cerebellar peduncles. L and R indicate, respectively, left and right.

**Figure 3 brainsci-13-01386-f003:**
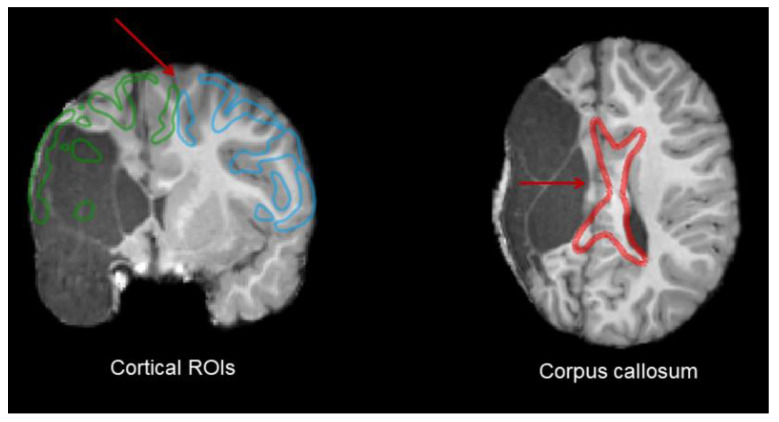
Misregistered cortical and corpus callosum ROIs for S6. In the left view, the green ROI represents the supposed ROI in the left hemisphere, and the blue is the right one. In the right view, the red delimitation represents the misregistered corpus callosum.

**Figure 4 brainsci-13-01386-f004:**
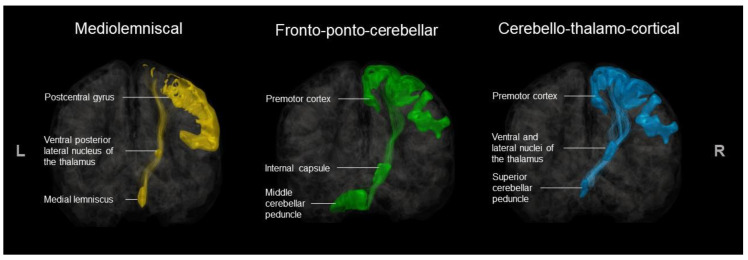
Included ROIs for ML, FPC, and CTF tract extraction.

**Figure 5 brainsci-13-01386-f005:**
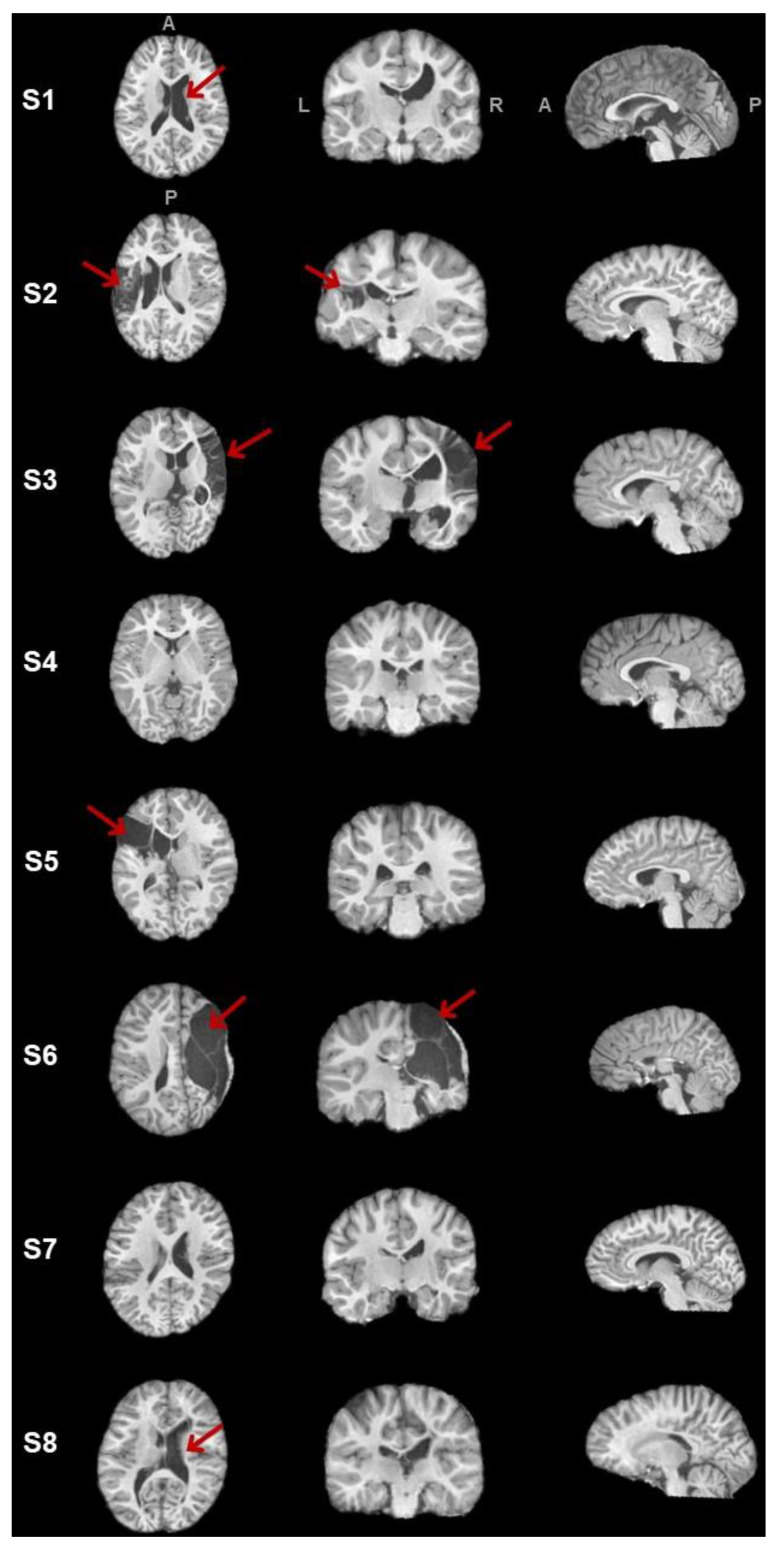
Lesion or abnormal structures (indicated with the red arrows) in axial, coronal, and sagittal views for participants. L and R stand for left and right, and A and P mean antero-posterior axis.

**Figure 6 brainsci-13-01386-f006:**
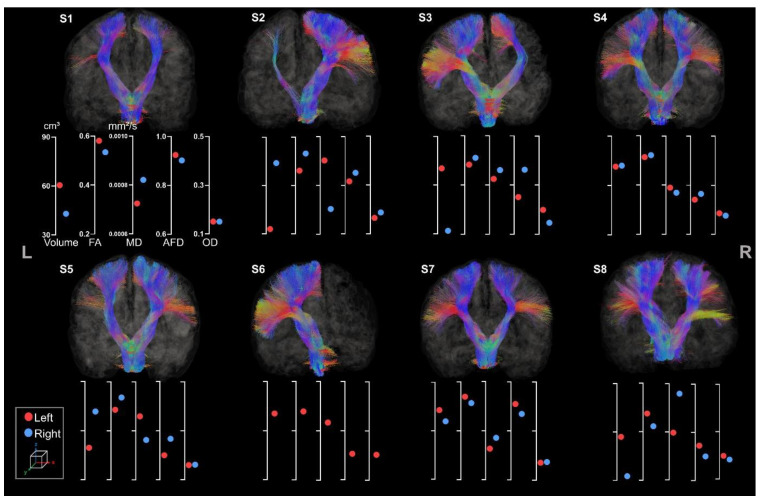
Coronal view of left and right CST for all participants. The left and the right side are indicated, respectively, with L and R. The first axis represents volume in cm^3^. The following represent DTI metrics in the following order: FA, MD, and fODF metrics: AFD and OD.

**Figure 7 brainsci-13-01386-f007:**
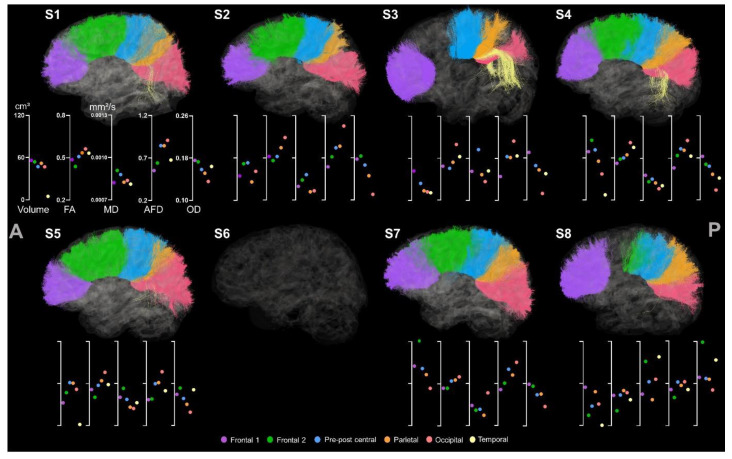
Sagittal view of the 6 segments of the CC for all participants. The anterior and posterior sides are indicated, respectively, with A and P. The first axis represents volume in cm^3^. The following represent DTI metrics in the following order: FA, MD, and fODF metrics: AFD and OD.

**Figure 8 brainsci-13-01386-f008:**
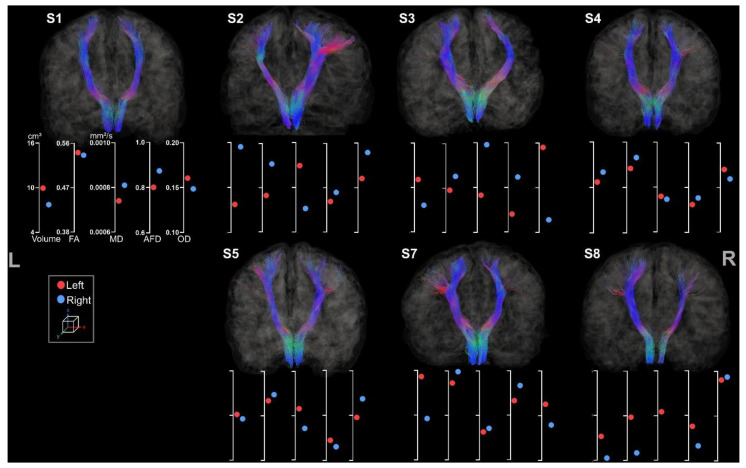
Coronal view of left and right ML tracts for the participants. The left and the right side are indicated, respectively, with L and R. The first axis represents volume in cm^3^. The following represent DTI metrics in the following order: FA, MD, and fODF metrics: AFD and OD.

**Figure 9 brainsci-13-01386-f009:**
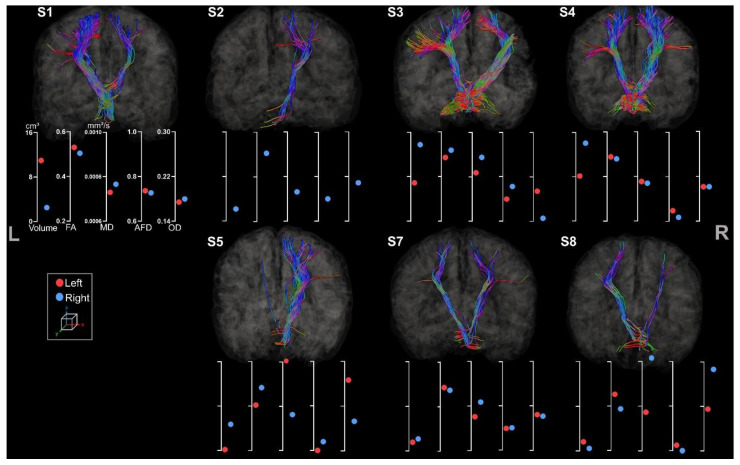
Tractography of right and left FPC tracts for the participants. For esthetical purposes, the tracts were visualized thanks to scil_visualize_bundles.py with --width 2 to improve the visualization. The first axis represents volume in cm^3^. The following represent DTI metrics in the following order: FA, MD, and fODF metrics: AFD and OD.

**Figure 10 brainsci-13-01386-f010:**
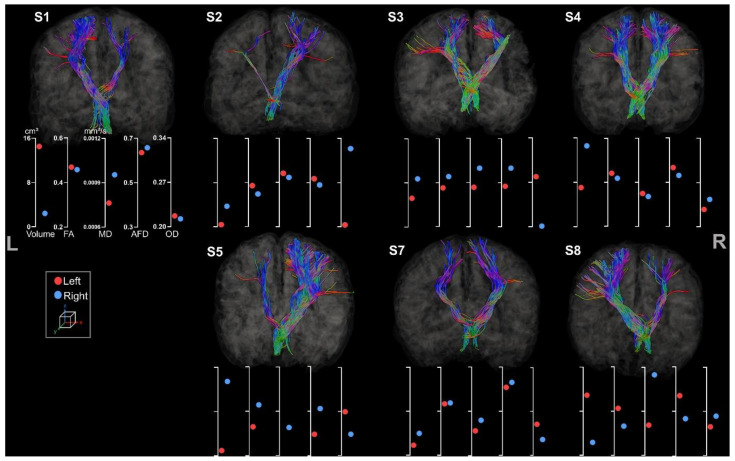
Tractography of right and left CTF tracts for the participants. For esthetical purposes, the tracts were visualized thanks to scil_visualize_bundles.py with --width 2 to improve the visualization. The first axis represents volume in cm^3^. The following represent DTI metrics in the following order: FA, MD, and fODF metrics: AFD and OD.

**Table 1 brainsci-13-01386-t001:** Demographic data of the participants.

	Sex	Age	Lesion Side	Etiology	MACS
S1	M	13 y, 10 m	Right	Pre- or perinatal stroke (undefined)	I
S2	M	12 y, 5 m	Left	Perinatal stroke	III
S3	F	8 y, 9 m	Right	Prenatal stroke	II
S4	M	9 y, 3 m	Right	Unknown	I
S5	F	11 y, 6 m	Left	Periventricular leukomalacia	II
S6	F	11 y, 4 m	Right	Prenatal stroke	III
S7	M	13 y, 3 m	Right	Prenatal stroke	I
S8	F	11 y, 3 m	Right	Prenatal stroke	II

**Table 2 brainsci-13-01386-t002:** Tissue volume (in cm^3^), tissue ratio, and AI for each participant.

	Left Hemisphere Volume		Right Hemisphere Volume				
	WM	GM	WM	GM	Brain Volume	Tissues Ratio	AI
S1	214.6	297	199.8	280.8	1166.1	1.06	0.03
S2	145.1	204.7	232.1	293.8	1075	0.67	0.2
S3	No segmentation						
S4	202.1	274.2	202.4	275.5	1131.5	1	0
S5	152.7	249.7	215.1	304.8	1088	0.77	0.13
S6	No segmentation						
S7	237.5	307.2	219.8	282.3	1217	1.08	0.04
S8	183.9	239	165	225.8	950.9	1.08	0.04

## Data Availability

Data are unavailable due to privacy or ethical restrictions.

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
