# Peer review of "The Challenge of Diffusion Magnetic Resonance Imaging in Cerebral Palsy: A Proposed Method to Identify White Matter Pathways"

_brainsci, 2023, doi:10.3390/brainsci13101386_

Round 1

Reviewer 1 Report

1-     Please provide a comprehensive detailed explanation of lines 131 to 133, including relevant information and findings for each case.

2-     The most intricate aspect of this study lies in effectively addressing the management of substantial lesional volumes. Therefore, it is imperative to offer a comprehensive elucidation of this challenge to ensure a thorough understanding.

3-     Provide a comprehensive and lucid description of the methodology employed to segment various components of the brain, encompassing both damaged and healthy tissues. If possible, include a graphical representation for enhanced clarity.

4-     Elaborate on the rationale behind employing a b-value equal to 300 in conjunction with the b-value of 1000 as mentioned in lines 139 and 140. Furthermore, it would be valuable to clarify the specific model employed for quantitatively extracting parameters.

5-     In reference to line 142 of the article, where AFD total, NuFO, and fixel AFD are mentioned, it is crucial to define AFD in its first usage. Furthermore, please specify the algorithm and software utilized to extract parameters for AFD total, NuFO, and fixel AFD.

6-     The discussion from lines 149 to 155 should encompass the challenges associated with segmenting the brain's white and gray matter regions and isolating them from affected areas. A comprehensive explanation of this aspect would enhance the understanding of the methodology.

7-     Please rectify the typographical error at the end of line 156 (S6), possibly intended as "Subject 6 (S6)."

8-     Elaborate on the atlas-based tailored procedure mentioned in line 171. It would be valuable to clarify how atlas-based techniques were employed, considering the deformations and variations in T1 images.

9-     Correct line 361 to "for S2's."

10-                        The figure and its caption pertaining to Figure 6 need revision to accurately reflect the view depicted. It is not a coronal view.

11-                        In the discussion and conclusion section, please elaborate on the supplementary advantages and options provided by the proposed pipeline in comparison to pipelines suitable for healthy subjects. Additionally, expounds on the primary advantages distinguishing the proposed methodology from tractography pipelines designed for healthy subjects. Additionally, highlight how these unique features are tailored to subjects with Cerebral Palsy, including their applicability to other lesioned brain conditions like tumors or strokes.

Reviewer 2 Report

The authors performed DTI in pediatric patients with CP. They used an existing, open-source data processing pipeline. The challenge in CP patients is data analysis like segmentation and fiber tracking in and around the affected brain regions.

The number of participants is not very big, however, it is not easy to recruit pediatric patients with CP. The overall scan time is under 30 min which limits the stress for the children. The authors describe the acquisition parameters and data processing procedures in great detail. The post-processing pipeline worked in 7 out of 8 patients and failed in the patient with a very large lesion. The reconstructed tracts and volumes show the asymmetry between affected and contralateral side. As expected, the NODDI and fODF metrics help to clarify misleading DTI results, since these more sophisticated methods deal with the problem arising from kissing/crossing fibers in the brain.

In conclusion, the manuscript describes the application of the data analysis pipeline in a small cohort of CP patients with heterogeneous brain structures.

Major comments:

The list of references really needs to be revised. First of all, in my opinion, over 100 references is too much for a research paper. The style of references does not follow the MDPI guidelines. Sometimes all authors are mentioned, sometimes only the first author + et al. The name of the journal is often missing, e.g., for references 11,17,22,58.... The name of D. Le Bihan (one of the most important researchers in the field) is not written correctly in Ref. 21 and 24. I have not checked all 105 references in detail, but, for example, in Ref. 82 and 84, there is only first author, title and year stated, nothing more. And check Ref. 35,39,40,61, there is something wrong with the titles.

Minor comments:

The volumes in Table 2 are presented with too many valid digits. The measurement is for sure not so precise.

Figs. 5 to 9:

The font size on the bars for the DTI etc. metrics is too small. Please write in the figure captions what the bars are showing. In Fig. 6, a sagittal view is presented, not a coronal view, please correct.

Round 2

Reviewer 1 Report

fine